# Elevated RBP4 in Subclinical Ketosis Cows Inhibits Follicular Granulosa Cell Proliferation and Steroid Hormone Synthesis

**DOI:** 10.3390/ani14213118

**Published:** 2024-10-29

**Authors:** Chang Zhao, Ruru Xu, Weizhe Yan, Benzheng Jiang, Shibin Feng, Xichun Wang, Hongyan Ding

**Affiliations:** 1College of Animal Science and Technology, Anhui Agricultural University, Hefei 230036, China; chang_zhao@ahau.edu.cn (C.Z.); xruru2019310040@163.com (R.X.); wolf1368755106@163.com (W.Y.); jbz19991012@126.com (B.J.); luyifsb@126.com (S.F.); 2Anhui Province Key Laboratory of Livestock and Poultry Product Safety Engineering, Institute of Animal Science and Veterinary Medicine, Anhui Academy of Agricultural Sciences, Hefei 230031, China

**Keywords:** dairy cow, retinol-binding protein 4, granulosa cells, PI3K/AKT pathway, follicular development

## Abstract

Our experiment detected RBP4 levels in the serum, liver, and ovaries of anestrus cows caused by subclinical ketosis. Compared with healthy estrus cows, the RBP4 levels were upregulated. Experiments were conducted using bovine granulosa cells (GCs) in vitro. The results indicate that the elevated RBP4 in SCK cows impacts GC proliferation, apoptosis, and steroid hormone synthesis through the STRA6 receptor and the PI3K/AKT pathway.

## 1. Introduction

During the periparturient period, dairy cows undergo an augmentation in milk production; however, their dry matter intake (DMI) fails to match this heightened demand. Consequently, their bodies mobilize stored fat to fulfill energy requirements, resulting in a state of negative energy balance (NEB) [1]. This metabolic state can subsequently induce subclinical ketosis (SCK) in dairy cows. Throughout the follicle development process, the proliferation and apoptosis of follicle granulosa cells (GCs) assume a pivotal role in sustaining oocyte growth and ovarian function [2,3,4]. The glucose (Glu) uptake and metabolism by GCs provide essential energy substrates and intermediates for oocyte maturation. In the presence of SCK in dairy cows, elevated levels of beta-hydroxybutyric acid (BHBA) and Glu levels can disrupt the metabolic functions of GCs. This disruption may lead to abnormal follicle development, influencing hormone secretion and synthesis and thereby adversely affecting reproductive performance. This impact extends to factors such as the onset of the first estrus, conception rate, and estrus cycle [5,6,7]. Although timed artificial insemination (TAI) techniques are widely used in cattle farms, early investigative studies have revealed that, in the early lactation period, SCK cows treated with TAI may still experience decreased reproductive capacity due to the absence of dominant ovarian follicles for ovulation [8]. This has an impact on the economic efficiency of the dairy industry and has become one of the significant risk factors affecting the reproductive health and metabolism of dairy cows [9].

Retinol-Binding Protein 4 (RBP4) is a carrier protein associated with lipid metabolism. Its primary function is to transport retinol from the liver and adipose tissue to the bloodstream and other parts of the body, where it performs a corresponding role. RBP4 is implicated in glucose and lipid metabolism in the liver and adipose tissue of cows [10]. Moreover, RBP4 plays a crucial role in significant reproductive biological processes in mammals, including steroid hormone production in the ovaries, oocyte maturation, and early embryonic development [11,12,13]. In current human medical research, it has been discovered that RBP4 is closely associated with the development of various metabolic diseases, such as common obesity and diabetes. Within the human body, RBP4 primarily mitigates insulin resistance by triggering the body’s inflammatory response [14]. In the pancreatic islet beta cells of rats, the RBP4 receptor STRA6 can modulate the RBP4-induced inhibition of insulin synthesis through the Janus kinase 2/STAT1/ISL-1 pathway [15]. This, consequently, effectively decreases circulating RBP4 levels and contributes to the recovery of and improvement in hyperglycemia in the bloodstream [16]. However, the association between RBP4 and postpartum anestrus in cows remains unclear. Building on our earlier proteomic findings, we observed elevated expression levels of RBP4 in the serum and follicular fluid of SCK anestrus cows. Additionally, by conducting pathway analysis on differentially expressed proteins in the serum and follicular fluid of SCK anestrus cows, we identified the involvement of the PI3K/AKT pathway. This pathway plays a crucial role in the energy metabolism of cows and mammalian follicle development and estrus [17,18]. Therefore, we hypothesize that the alterations in RBP4 in SCK cows may be intricately linked to postpartum follicular development.

The PI3K/AKT pathway is a classical insulin metabolic pathway, and in instances of insufficient postpartum cow nutrition and energy, it frequently coexists with inflammation and metabolic issues [19,20]. This pathway plays a pivotal role in diverse cellular responses [21], including cell proliferation, differentiation, apoptosis, and Glu transport. AKT, serving as a signaling molecule, facilitates the transmission of signals that initiate the PI3K pathway. Once activated, AKT stimulates the activation of glucose transporter-4 (GLUT4), promoting the cellular uptake of glucose. The PI3K/AKT pathway plays a crucial role in energy metabolism [22]. Furthermore, a growing body of experiments has revealed that the PI3K-AKT signal can regulate ovarian function, encompassing the recruitment of primordial follicles, granulosa cell proliferation, corpus luteum development, and oocyte maturation [23]. It also plays a crucial role in follicle development and steroid production [24]. Therefore, it is evident that PI3K/AKT plays a pivotal role in the proliferation of cow follicular cells. Simultaneously, based on the findings from our earlier proteomic studies, it was observed that RBP4 is a differentially expressed protein in the follicular fluid of cows experiencing NEB and those of healthy dairy cows, with the PI3K/AKT pathway being one of the enriched pathways [22]. An increase in the milk yield of dairy cows causes an increase in the incidence of SCK. The pathogenesis of postpartum reproductive disorders caused by SCK has become a hot research topic of interest. Based on previous findings, the objective of this study is to examine how RBP4 influences the proliferation, apoptosis, and steroid hormone secretion in GCs of SCK cows through the PI3K/AKT pathway. This investigation establishes the groundwork for exploring the connection between SCK and postpartum follicle development in cows and clarifies the mechanism by which the SCK-induced elevation of RBP4 inhibits the function of GCs.

## 2. Material and Methods

### 2.1. Animals

Animal grouping was conducted in compliance with the guidelines established by the Ethics Committee of the Chinese Ministry of Agriculture; this experiment was carried out at an intensive dairy farm in Heilongjiang Province, China. The management of the Total Mixed Ration (TMR) strictly adhered to the Chinese standards for lactating dairy cow management. The animals in the experiment were treated in accordance with the guidelines approved by the Animal Welfare and Research Ethics Committee of Anhui Agricultural University, China (Approval Number: 2021012-1). Milking of lactating cows was conducted at 06:30, 12:00, and 19:30 daily. The composition and chemical components of the TMR feed are detailed in Appendix A.

We randomly selected similar lactation, body condition score (BCS), parity, and age healthy Holstein cows (median number of lactations = 3, range = 2–4; daily average milk production at day 60 postpartum: <35 kg/d = 29 cows, ≥35 kg/d = 89 cows; body condition score (BCS): median = 3.00, range = 2.75–3.25). Based on 1.20 mM ≤ BHBA < 2.00 mM, from day 14 to 21, postpartum cows were assigned to the SCK group (n = 30), and BHBA < 1.20 mM were selected as the control group [25,26]. The cows were then monitored further until the postpartum period of day 45 to 60. During this stage, none of the cows involved in this experiment received hormonal ovulation treatments. Between day 45 and 60 postpartum, clinical veterinarians excluded cows with other diseases or clinical abnormalities based on serum indicators and clinical symptoms. Ovarian development, corpus luteum status, and follicular development were assessed through rectal examination and ultrasound examination (using equipment from the Honda company, Tokyo, Japan). Based on these evaluations, cows that exhibited anestrus were selected for the SCK-AE group (n = 12), characterized by the absence of dominant follicles, a maximum follicle diameter less than 8 mm, no corpora lutea on the ovaries, and no clinical signs of estrus. Additionally, cows with normal estrus were selected for the C-E group (n = 12), characterized by dominant follicles larger than 8 mm and clinical signs of estrus [27]. Basic information about the SCK-AE and C-E group cows is provided in Appendix A.

### 2.2. Sample Collection

We collected blood samples via the caudal vessels before morning feeding at 14–21 days and 45–60 days. After centrifugation of the blood at 1500× *g* for 5 min at 4 °C, we collected the supernatant and centrifuged it again at 12,000× *g* for 5 min at 4 °C to separate serum. The resultant supernatant was stored at −80 °C.

The experimental cows underwent a 12 h fasting period prior to electric shock slaughter performed using a handheld hemp appliance (BFSD-001, Luxin Qida, Weifang, China). Both ends of the handheld hemp appliance were dipped in a 5% NaCl solution to enhance conductivity. Subsequently, one end of the electrode was firmly pressed against the junction of the cow’s eye and ear root at 90 V for 5–8 s. The cows were bled after death. Disinfected surgical knives were utilized to incise the abdominal cavity of cows and extract both the left and right liver lobes, as well as the ovaries. The tissues were washed using 9% NaCl solution in a sterile environment. Parts of the tissue samples were stored at 80 °C after quick freezing in liquid nitrogen, and the remaining tissue was stored in paraformaldehyde. Liver and ovarian tissues (50 mg each) were added with total protein extraction solution (lysate–protease inhibitor–phosphatase inhibitor = 100:1:3) at a ratio of 1:10 (g/mL), and the tissue was cut up and homogenized using an electric homogenizer. Thereafter, the tissue samples were incubated for 20 min at 4 °C in a refrigerator to allow sufficient lysis and centrifuged at 12,000× *g* for 10 min at 4 °C, and the supernatant was removed, placed in a new centrifuge tube, and stored at 20 °C.

### 2.3. Isolation and Culturing of Bovine Granulosa Cells

We conducted isolation and culture of bovine granulosa cells based on other studies [28,29,30]. We acquired cow ovaries (no corpus luteum on the ovary) from the slaughterhouse (Anhui and Heilongjiang, China) and returned them to the laboratory within 1 h. We washed the ovaries three times in a 0.9% physiological saline solution (Gibco, Waltham, MA, USA; 10099-141) supplemented with penicillin and streptomycin. We used an 18-gauge needle to extract the follicular fluid from follicles with diameters ranging from 2 to 6 mm and collected it in 15 mL centrifuge tubes for GC isolation. We filtered the aspirated follicular fluid using a 40 μm filter to eliminate oocyte–cumulus complexes and cellular debris. We then centrifuged the liquid containing the GCs twice at 100× *g* for 5 min each time. We lysed red blood cells by resuspending the GC pellets in 1× red blood cell lysis buffer, which was terminated by the addition of Dulbecco’s modified Eagle’s medium (DMEM)/F-12 (SparkJade, Jinan, China, CD0001) medium containing 10% fetal bovine serum (FBS, ExCell, Suzhou, China, FSP500). We grew the cells in DMEM/-12 (SparkJade) medium supplemented with 10% FBS (ExCell) and 1% double antibiotics (Gibco Hyclone) at 37 °C, 5% CO_2_, and saturated humidity, at a density of around 5 × 10^5^ cells/mL. Every 48 h during incubation, we changed the medium until 80 to 90% confluency was obtained. All treatment groups were supplemented with 0.1 µM androstenedione (CLOUD-CLONE Corp., Wuhan, China), which is a precursor of estrogen synthesis.

### 2.4. Detection of Blood Biochemical Indices

We used BHBA serum ketone tester and corresponding test strips (Yicheng, Beijing, China: TBS-1) to determine the concentration of BHBA in cow serum. The measurement of Glu in serum was carried out using a biochemical analyzer, and the relevant reagent kit was purchased from Nanjing Jiancheng Bioengineering Research Institute (F006-1-1, Nanjing, China). The specific operation method was carried out according to the instructions provided by the merchant.

### 2.5. Calculating the Growth Rate of Follicles

A B-ultrasound device was used to check the diameter of the largest follicles on both ovaries at 45 and 60 days postpartum. We calculated the rate of follicular development over these 15 days, and the formula for calculating the daily follicular development rate was as follows:∆LF = (LF2 − LF1)/15

∆LF represents the maximum follicle growth rate (mm/day) from day 45 to 60 postpartum, and LF2 and LF1 represent the diameters (mm) of the largest follicles at day 45 and 60 postpartum, respectively.

### 2.6. Western Blot Assay

Treated tissue or cell samples were treated with 1% protease inhibitor (PMSF; Beyotime, Shanghai, China) in lysis buffer (RIPA; Beyotime, Shanghai, China) cracked on ice. The lysate was centrifuged at 4 °C, 15,000× *g* for 15 min, and the protein concentration was determined with a BCA kit (Beyotime, Shanghai, China). Samples containing 50 µg of protein were separated using a 12% sodium dodecyl sulfate–polyacrylamide gel separation (SDS-PAG, Servicebio, Wuhan, China), and the excised protein of interest gel was transferred to a PVDF membrane (Servicebio, Wuhan, China) [31,32].

Membranes were blocked by 5% skimmed milk for 2 h and then incubated with primary antibody (GAPDH, Servicebio, GB15002, Wuhan, China; StAR, Servicebio, GB111430, Wuhan China; HSD3B1, A19266, ABclonal, Boston, MA, USA; CYP27B1, Servicebio, GB114918, Wuhan, China; CYP17A1, Servicebio, GB112095, Wuhan, China; GLUT4, Servicebio, GB11217, Wuhan, China; Bax, Servicebio, GB114122, Wuhan, China; Caspase-9, Servicebio, GB12053, Wuhan, China; CDK1, Servicebio, GB11398, Wuhan China; CCNE2, Proteintech Group, 11935-1-AP, Chicago, IL, USA; CCNA1, Servicebio, GB113964, Wuhan, China; CCND1, Proteintech Group, 60186-1-LG, Chicago, IL, USA; AKT, Bioss, bs-5146R, Beijing, China; p-AKT, Bioss, bs-2720R, Beijing, China; RBP4, Cloud-Clone, MAA929Bo21, Wuhan, China) at 4 °C overnight. Goat anti-rabbit IgG, Affunit, s0008, was incubated with HRP-conjugated goat anti-rabbit secondary antibody (Affunit, s0008) for 45 min at 20–26 °C. Signals were detected with a chemiluminescent substrate kit (ECL, Servicebio, Wuhan, China), and grayscale analysis was performed using ImageJ software (1.41).

### 2.7. Treatment of GCs

RBP4 treatment: recombinant bovine RBP4 protein (Cloud clone Corp., RPA929Bo01, Wuhan, China, endotoxin content < 1.0 endotoxin units per 1 µg protein, size 22 kDa, purity > 95%) was diluted to 300 ng/mL, 200 ng/mL, and 100 ng/mL, added to GCs, and incubated for 12 h, 24 h, and 48 h, with three replicates in each group.

Silencing siSTRA6: The transfection experiment was performed according to the instructions provided by Tsingke Biotechnology (Nanjing, China). In total, 6 × 10^4^ cells were seeded in 400 μL antibiotic-free medium and cultured for 24 h, followed by the addition of siSTRA6 (Tsingke Biotechnology Co., Ltd., Nanjing, China). After 24 h of cell transfection and incubation, the culture medium was replaced with RBP4 containing medium, and incubation was continued for 48 h.

### 2.8. Cell Viability Assay

We conducted the cell viability test based on the research methods of others [33,34]. Following cell collection, viability testing was conducted using a Cell Counting Kit-8 (CCK-8) (Beyotime, Shanghai, China).

### 2.9. Total RNA Extraction, cDNA Synthesis, and RT-qPCR

Total RNA was extracted using TRizol (Life Technologies, New York, NY, USA) reagent according to the protocol. Genomic reactions were removed. RNA concentration was measured using a NanDrop 2000 spectrophotometer (Thermo Fisher Scientific, Waltham, MA, USA), calculating the ratio of UV activity at 260/280 nm to measure the purity of RNA. Randon and Olido-DT were mixed and centrifuged. Following the instructions, the mixture was incubated at 65 °C for 5 min in a fridge; then, dNTP, MDL, 0.1 M DTT, and 5 × buffer were added to the reaction system and mixed before centrifugation. The first step involved a 60 min reverse transcription at 42 °C in a water bath, followed by a 10 min incubation at 85 °C to deactivate the reverse transcriptase enzyme. After completing the reaction, the cDNA was stored at −20 °C for future use. β-actin and GAPDH were used as reference genes for all genetic tests. Target genes were designed and synthesized according to GenBank sequences, and the gene primer sequences used in the experiment are listed in Table 1. The real-time fluorescence quantitative polymerase chain reaction (RT-qPCR) reaction mixture was prepared following the instructions provided by the Novo Start SYBR qPCR Super Mix Plus kit (NovoProtein, E096, Shanghai, China). The RT-qPCR reaction conditions were set as follows: 95 °C for 2 min, followed by 40 cycles of 94 °C for 20 s, 60 °C for 20 s, and 72 °C for 30 s. The relative expression values were calculated using the 2^−ΔΔCt^ method.

### 2.10. Immunofluorescence

The prepared GCs were seeded onto glass slides and incubated with the primary FSHR antibody (FSHR-Ab, Affinity Biosciences, AF5242, Beijing, China) and FITC-labeled secondary antibody (goat anti-rabbit antibody, AffuUnit, s008). Subsequently, nuclear staining was performed using DAPI. After staining, cells were observed under a confocal microscope and photographed.

### 2.11. Cell Cycle Analysis

At the end of the GC grouping treatment, GCs were collected, incubated with 0.25% trypsin, and then washed with pre-cooled PBS. Then, 1 mL of pre-cooled 70% ethanol was added, and the cells were fixed at 4 °C for 2 h. The cells were collected by centrifugation at 1000× *g* for 5 min. Propidium Iodide (PI) staining solution was prepared according to the manufacturer’s instructions, and 0.5 mL of PI was added to each tube of cells. The GCs were then incubated in the dark at 37 °C for 30 min. Finally, red fluorescence at an excitation wavelength of 488 nm was detected using flow cytometry.

### 2.12. Cell Apoptosis Analysis

Flow cytometry was used to detect cell apoptosis. GCs were collected and washed twice with PBS. Each sample was mixed with 500 µL of 1 × Binding Buffer working solution and 10 µL of Annexin V-FITC. The samples were incubated for 20 min in the dark, centrifuged, and resuspended, followed by the addition of 5 µL of PI 5 min before the test.

### 2.13. Statistical Analysis

The data are presented as the mean ± SEM of at least three independent replicates. Statistical analysis was conducted using one-way analysis of variance (ANOVA) in SPSS 16.0 software (IBM Corp., Armonk, NY, USA), and graphs were generated using Graph Pad Prism 10.3.1 software (GraphPad Inc., La Jolla, CA, USA). Western blot results were analyzed using Image J 1.41 software, and flow cytometry results were analyzed and plotted using FlowJo-VX.

## 3. Results

### 3.1. Serum Biomarker and AKT Phosphorylation Level Assay in Dairy Cows

In order to investigate the effects of SCK on the energy metabolism in dairy cows, we measured the concentrations of BHBA, Glu, and RBP4 in the serum of cows from both the C-E group and the SCK-AE group at day 14 to 21 postpartum. As shown in Figure 1, at day 14 to 21 postpartum, compared to the C-E group, SCK-AE group cows exhibited a significant decrease in serum Glu concentration (*p* < 0.01) and a significant increase in BHBA and RBP4 concentration (*p* < 0.01). In order to compare the protein expression differences of RBP4 and key factors in the glucose metabolism pathway, AKT and p-AKT, in the liver and ovaries of C-E and SCK-AE group cows, we used the WB method to detect these parameters. As shown in Figure 1D–F, in both the liver and ovaries, the expression of the RBP4 protein was significantly higher in the SCK-AE group than in the C-E group (*p* < 0.05), and the expression of the p-AKT protein was significantly higher in the SCK-AE group (*p* < 0.05), while the expression of AKT showed no significant difference between the two groups (*p* > 0.05) (Figure 1G–I).

### 3.2. The Impact of RBP4 on the Activity and Function of Bovine GCs

Immunofluorescence was used to detect the expression of FSHR in GCs. As shown in Figure 2A, the GCs were positive for FSHR and could be used for subsequent tests. This study revealed that in both the liver and ovaries, the expression of the RBP4 protein was significantly higher in the SCK-AE group than in the C-E group (*p* < 0.05). To further investigate the impact of RBP4 on follicular development in the ovaries, we isolated primary bovine ovarian GCs in vitro and stimulated GCs with different doses of exogenous recombinant RBP4 protein (100, 200, 300 ng/mL) for varying durations (12, 24, 48 h). We found that RBP4 inhibited the cell viability of primary bovine GCs in a dose- and time-dependent manner. Under the conditions of 48 h treatment with 100, 200, and 300 ng/mL, the cell viabilities were 80.6%, 54%, and 46.8%, respectively. Among these, the stimulation with 200 ng/mL for 48 h resulted in nearly half of the cells becoming non-viable, which was used for subsequent experiments (Figure 2B).

To further elucidate the detrimental effects of RBP4 on GCs, we assessed cell proliferation and apoptosis in GCs after 48 h of stimulation with different concentrations. Compared to the control group, the RBP4-200 ng/mL and RBP4-300 ng/mL groups showed a highly significant increase in the proportion of G1-phase GCs (*p* < 0.01). In contrast, the proportions of S-phase GCs at all three concentrations significantly decreased (*p* < 0.01). These results indicate that RBP4-200 ng/mL and RBP4-300 ng/mL induced cell cycle arrest at the G1 phase (Figure 3A–E). Therefore, the significant interruption of the cell cycle may be associated with the reduced proliferation of GCs. Flow cytometry was used to detect apoptosis in GCs. Compared with the control group, the apoptosis rate of GCs increased with the increase in RBP4 addition concentration (Figure 3I–L). We used the WB and RT-qPCR methods to assess the protein levels and relative mRNA expression of cell cycle proteins CDK1, CCNE2, CCND1, and CCNA1. Compared to the control group with 0 ng/mL, all three concentrations of RBP4 stimulation (100, 200, 300 ng/mL) significantly reduced the mRNA content and protein expression levels of CDK1, CCNE2, CCND1, and CCNA1 in bovine GCs (*p* < 0.01) (Figure 3F–H). Compared with the control group, the expression levels of steroid secretion related proteins (StAR, HSD3B1, CYP27B1, and CYP17A1) in GCs significantly decreased after treatment with 300 ng/mL RBP4 (Figure 3T,U), and the secretion levels of E_2_ and P_4_ in the GC culture medium significantly decreased when treated with 300 ng/mL RBP4 (Figure 3R,S).

To further assess whether RBP4 affects cell apoptosis in addition to its impact on GC proliferation, we measured the number of viable cells under different concentrations of RBP4 stimulation. Compared to the 8.2% cell apoptosis rate with 0 ng/mL, the GCs’ apoptosis rates were significantly increased under RBP4 stimulation at 100, 200, and 300 ng/mL, with rates of 12.1%, 13.8%, and 16%, respectively (*p* < 0.01) (Figure 3I–M). To further confirm our findings, we used the WB and qPCR methods to assess the expression of apoptosis genes, Bax and Caspase-9. As the RBP4 concentration increased, it significantly elevated the gene and protein levels of Bax and Caspase-9 (*p* < 0.01) (Figure 3N–P).

Furthermore, we also investigated the impact of RBP4 on the secretion and synthesis of steroid hormones in GCs. The experimental results revealed that under the stimulation of RBP4, the expressions of the GC-specific receptor FSHR and essential enzymes in the process of steroid hormone secretion, such as StAR and HSD3B1, were all suppressed (Figure 4A–C). Regarding the mRNA and protein levels, different concentrations of RBP4 were found to reduce the levels of StAR, HSD3B1, CYP17A1, and CYP27B1, which are associated with the secretion and synthesis of steroid hormones (Figure 4G–I). Glu is an important metabolic substrate, and GCs utilize glucose to provide energy for oocytes. To analyze whether RBP4 affects the glycolytic pathway in bovine GCs, we examined the expression of GLUT4 and key factors in the signaling pathway, AKT and p-AKT. The results showed that as the concentration of RBP4 stimulation in GCs increased, GLUT4 protein activity decreased, and it enhanced the phosphorylation of AKT (Figure 4D–F).

### 3.3. RBP4 Activates STRA6 to Inhibit GC Function via the PI3K/AKT Pathway

To further elucidate the molecular mechanism of RBP4 inhibiting follicular development, we attempted to investigate whether RBP4’s membrane receptor, STRA6, is involved in this process. We first detected the expression of STRA6 in the liver, ovaries, and GCs using WB and immunofluorescence. We found that STRA6 is expressed in all of these tissues, and the expression of STRA6 in SCK cows is significantly higher compared to the normal control group (*p* < 0.05) (Figure 5A–D). At the same time, we observed that RBP4 can dose-dependently increase the protein expression of STRA6 (Figure 5E,F).

We transfected siRNA specifically inhibiting STRA6 into GCs in the presence of added RBP4. Transfection was performed at a concentration of 50 nM for 24 h, followed by a 48 h treatment with RBP4. When compared to the group treated with RBP4 alone, the results showed that siRNA transfection effectively blocked the RBP4-induced upregulation of STRA6 (Figure 6A,B). This resulted in an increase in GLUT4 expression (Figure 6F,G), which enhanced upregulation in the expression of proteins related to steroid hormone secretion and synthesis, including StAR, HSD3B1, CYP17A1, and CYP27B1 (Figure 6C–E). In addition, it led to promoted cell proliferation (Figure 7A,B) and relieved cell apoptomy (Figure 7C–G).

By using the PI3K/AKT pathway activator SC79, which effectively activates *p*-AKT expression (Figure 8C), GCs were divided into four groups and cultured for 24 h. The selected inhibitor concentration was 10 μM. The results demonstrated that the activation of AKT effectively prevented the proliferative inhibition caused by RBP4 (Figure 8A,B,D,E), reduced cell apoptosis (Figure 8F–H), and promoted steroid hormone protein expression (Figure 9A–C) as well as glucose metabolic pathways (Figure 9D,E).

## 4. Discussion

In the periparturient period, dairy cows frequently undergo a state of negative energy balance, potentially leading to the development of SCK. This condition includes heightened fat mobilization, suppressed glucose metabolism, diminished insulin sensitivity, and the onset of insulin resistance [35]. The liver is a pivotal organ for energy metabolism in the body, and research has been conducted on the role of RBP4 in the liver. For instance, injecting purified RBP4 or RBP4 obtained through transgenic overexpression into mice activates hepatic phosphoenolpyruvate carboxykinase 1 (PCK1), promoting gluconeogenesis in the organism and diminishing insulin sensitivity in mouse muscles; the result indicates the participation of RBP4 in hepatic glucose metabolism [11]. The ovary contains numerous ovarian follicles of varying sizes, and within these follicles, the oocyte not only requires energy for its own growth and development but also contributes to the energy supply for the growth and development of the embryo [10]. Hence, energy metabolism within the follicles plays a crucial role in ovulation and embryonic development. Obese women with polycystic ovary syndrome exhibit higher concentrations of RBP4 in serum compared to normal individuals, often accompanied by features of insulin resistance [36]; the overexpression of RBP4 inhibits the growth and development of pig granulosa cells while promoting apoptosis [17]. However, research on RBP4 in the bodies of cows is relatively limited. According to our experimental findings, the expression levels of RBP4 and p-AKT in the liver and ovaries of SCK group cows increased, indicating that elevated concentrations of RBP4 influence the P13K/AKT pathway.

GCs play a crucial role in the process of follicular development. The growth and development of GCs involve multiple biological processes and are subject to various complex signal regulations, which also impact the maturation of oocytes. Therefore, the growth status of GCs is crucial for follicular development. Many studies have shown that RBP4 can be found in the GCs of ovaries in various animals, including mice [11], pigs [37], and cows [38]. In this experiment, we first performed immunofluorescence identification on the primary isolated bovine GCs and found that FSHR was expressed, proving that the model was successfully established and can be used for subsequent experiments. Further stimulation of GCs with RBP4 revealed that RBP4 inhibits the GCs’ activity in a dose- and time-dependent manner. Additionally, it impairs the GCs’ functionality, leading to the suppression of steroid hormone secretion and synthesis. Regarding the impact of RBP4 on steroid hormones [39], Wang Qi and colleagues found in their study on the testes of Bactrian camels that the overexpression of RBP4 inhibits the transcription and translation of steroidogenic enzymes HSD3B1 and SRD5A1 in testicular supporting cells; this results in decreased levels of dihydrotestosterone and androgen receptor expression. The continued deletion of RBP4 alleviates this inhibitory effect to some extent, and this is somewhat similar to our findings. The proliferation and apoptosis of GCs have a significant impact on follicle development. The experimental results of Huang Rong and others indicate that RBP4 can regulate the proliferation, migration, and apoptosis of ESCs, affecting the receptivity of the uterine endometrium in pigs, thereby influencing embryo implantation [40]. Studies have found that the overexpression of RBP4 in ovarian cancer cells promotes the proliferation and migration of cancer cells [38]. These studies suggest that RBP4 plays a role in both the proliferation and apoptosis of normal and cancer cells. We detected the expression levels of Bax, Caspase-9, CDK1, CCNE2, CCND1, and CCNA1, which are related to apoptosis and the cell cycle. Flow cytometry was used to assess the distribution of cells in different cell cycle phases. Our findings indicate that RBP4 stimulation inhibits the proliferation of GCs and induces cell apoptosis. GLUT4 regulates intracellular energy metabolism and functional balance by mediating the entry of glucose into cells, thereby influencing whole-body metabolism [39]. Compared to healthy women, overweight women with polycystic ovary syndrome (PCOS) have an increased expression of RBP4 in adipose tissue and adipocytes, with a decreased GLUT4 expression [14]. Studies have shown that elevated RBP4 levels in serum, in both humans and animals, may lead to the downregulation of GLUT4 in adipocytes, resulting in insulin resistance [41,42]. In our research, we found that adding RBP4 to bovine GCs led to a decrease in GLUT4 expression, and as the RBP4 concentration increased, the GLUT4 expression decreased. Although RBP4 expression in the ovaries is lower than in the liver, it still affects the homeostasis of GLUT4 and disrupts the body’s energy balance.

STRA6 is the specific receptor for RBP4 and plays a role in transporting retinol into cells through the cell membrane. In addition to serving as a carrier for retinol, STRA6 also acts as a cell surface signaling receptor for RBP4, influencing the occurrence of intestinal tumors and insulin secretion in pancreatic beta cells [15,43]. In this study, we detected the expression of STRA6 in bovine GCs for the first time using immunofluorescence, suggesting that RBP4 might exert its effects on GCs through the mediation of STRA6. Nevertheless, the specific mechanisms require further validation. The experiments revealed that RBP4 upregulates the expression of the STRA6 receptor. However, it was observed that silencing the STRA6 receptor effectively alleviated RBP4-induced phenomena such as GC apoptosis, disrupted glucose metabolism, AKT phosphorylation, decreased steroid hormone secretion, and reversed impaired cell proliferation. These findings further support the role of RBP4 in affecting GCs through the STRA6 receptor [15]. The above results indicate the importance of STRA6 in insulin secretion, its involvement in insulin resistance and type II diabetes, and its role in regulating retinol intake and metabolism in mice under the influence of FSH.

The PI3K/AKT pathway is a classical insulin signaling pathway. Postpartum cows experience NEB, and it is often accompanied by inflammatory reactions and metabolic issues. The PI3K/AKT pathway plays a crucial role in various cellular responses, including cell proliferation, differentiation, apoptosis, and glucose transport [22,31]. Moreover, the PI3K/AKT signaling pathway also plays a crucial role in follicle development and steroidogenesis. During the process of follicle development, there is a requirement for Glu breakdown functionality. Glu, as a vital fuel for metabolism, exerts its effects through the PI3K/AKT signaling pathway. Studies have found that AKT can regulate the uptake and utilization of Glu and amino acids by bovine mammary gland cells, thereby increasing energy supply to mammary cells and milk production. Hence, PI3K/AKT plays a significant role in energy metabolism [23,26,44]. Research indicates that AKT plays a crucial regulatory role in follicle development by participating in the modulation of gene expression levels associated with follicle development and corpus luteum formation. Through the activation of downstream signaling molecules, AKT influences the proliferation, differentiation, and apoptosis of follicle cells, serving as a key regulatory factor in follicle development. GLUT4, as a membrane protein associated with Glu transport, is responsible for regulating the uptake and utilization of Glu [45]. The PI3K/AKT pathway plays a crucial role in the regulation of GLUT4. Research indicates that when the PI3K/AKT pathway is activated, AKT directly phosphorylates and activates multiple downstream factors, including acetyl-CoA carboxylase and TBC1D4 (TBC1 Domain Family Member 4); these factors can promote the translocation of GLUT4 to the cell membrane surface. Through in vivo experiments, we observed that the p-AKT levels in SCK-AE cows were higher than those in the C-E group cows. In vitro stimulation of GCs with RBP4 recombinant protein revealed that with an increase in RBP4 concentration, the phosphorylation level of AKT also increased, leading to the inhibition of GLUT4 expression. The involvement of RBP4 in the glucose metabolism process of GCs is evident. To further investigate the role of the PI3K/AKT pathway in the influence of RBP4 on GCs’ glucose metabolism, we added the activator SC79 to facilitate the phosphorylation of AKT. The results revealed that this effectively alleviated the proliferation inhibition of GCs caused by RBP4 stimulation, as well as the inhibition of GLUT4. It is evident that RBP4 inhibits the normal glucose metabolism function of GCs by suppressing the PI3K/AKT pathway, thereby affecting the transport of GLUT4. In this context, we have demonstrated that RBP4-STRA6 can activate the PI3K/AKT pathway, influencing Glu transport. While we only measured the levels of GLUT4, it cannot be ruled out that RBP4 may have other potential effects on GCs. Further investigation is needed in this regard. Additionally, we explored the role of the PI3K/AKT pathway in the influence of RBP4 on steroid hormones in GCs. The PI3K/AKT pathway plays a crucial role in the regulation of steroid hormone synthesis. As is well known, FSH can influence the granulosa cells’ cell cycle and mitosis by activating the PI3K/AKT pathway. In bovine testicular cells (TCs), luteinizing hormone (LH) has been shown to stimulate the production of androgens and the expression of CYP17A1 through the activation of the PI3K/AKT pathway [46]. Studies on TCs in goat follicles have also revealed that the PI3K/AKT pathway regulates the synthesis of steroids in TCs [47]. The above research results and the results of this experiment both indicate that the PI3K/AKT pathway can influence the metabolism and synthesis of steroid hormones by regulating the enzymatic activity of StAR, HSD3B1, CYP17A1, and CYP27B1. The activator of AKT SC79 could reverse the RBP4-induced inhibition of StAR, HSD3B1, CYP17A1, and CYP27B1 expression in GCs. In our study, we found that the PI3K/AKT pathway increased the expression of steroid-hormone-secretion-related proteins and genes while promoting the secretion of E_2_. Therefore, the impaired follicle development in cows suggests that further research is needed to investigate the cellular responses stimulated by RBP4-STRA6.

## 5. Conclusions

SCK leads to an elevation in the concentration of RBP4 in anestrus cows. The increased RBP4 influences the synthesis of steroid hormones and the proliferation and apoptosis of GCs through the STRA6/PI3K/AKT pathway, thereby impacting follicular development in SCK cows (Figure 10).

## Figures and Tables

**Figure 1 animals-14-03118-f001:**
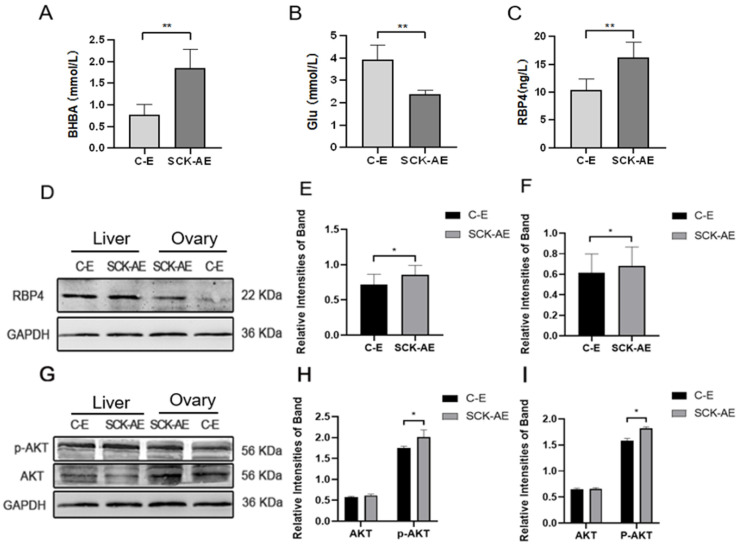
Comparison of postpartum day 14 to 21 related parameters in C-E group (n = 12) and SCK-AE group (n = 12) cows; (**A**) concentration of BHBA in the serum; (**B**) concentration of Glu in the serum; (**C**) concentration of RBP4 in the serum; (**D**) immunoblot results for RBP4 in the liver and ovarian tissues of both groups; (**E**) protein expression levels of RBP4 in the liver; (**F**) protein expression levels of RBP4 in the ovaries; (**G**) immunoblot results for p-AKT and AKT in the liver and ovaries; (**H**) protein expression levels of p-AKT and AKT in the liver; (**I**) protein expression levels of p-AKT and AKT in the ovaries. * *p* < 0.05. ** *p* < 0.01 indicates significant difference between the two groups (*p* < 0.01).

**Figure 2 animals-14-03118-f002:**
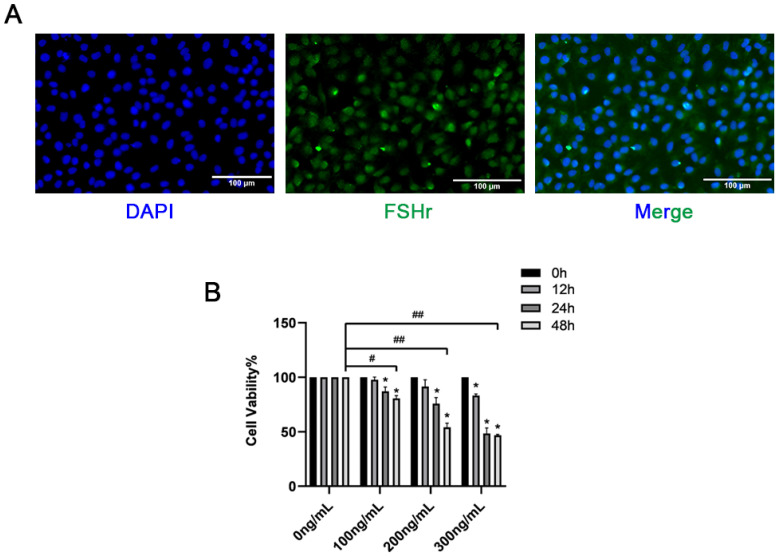
Identification of GCs and screening of RBP4 concentration; (**A**) immunofluorescent identification of FSHR expression in bovine GCs; (**B**) inhibition of cell viability by RBP4. ^##^ indicates extreme significance (*p* < 0.01); ^#^ indicates significance (* *p* < 0.05).

**Figure 3 animals-14-03118-f003:**
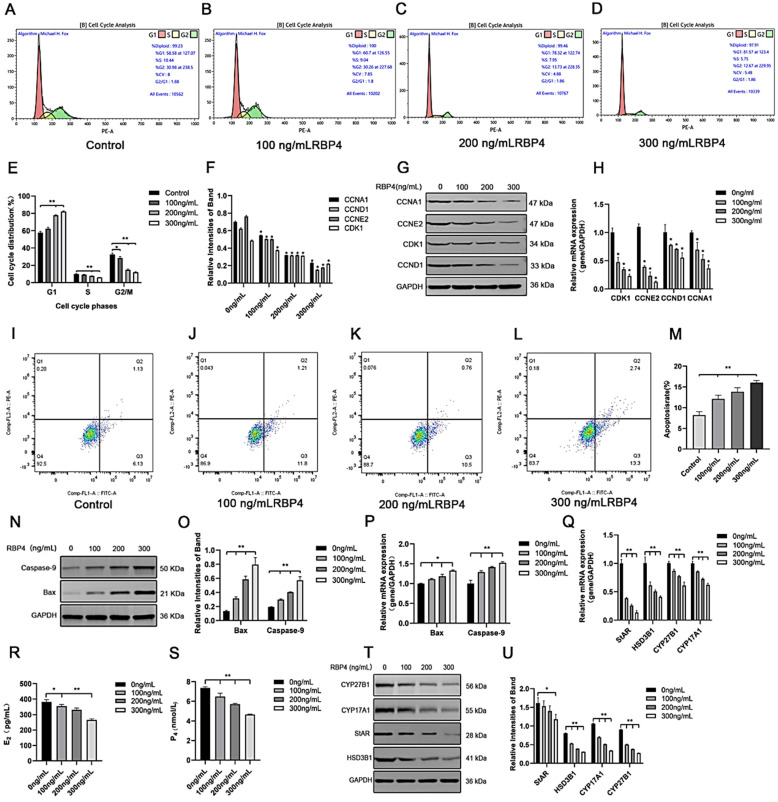
The inhibitory effect of exogenous RBP4 on the activity and function of GCs; (**A**) flow cytometric analysis of the cell cycle distribution of GCs in the control group (0 ng/mL RBP4); (**B**) flow cytometric analysis of the cell cycle distribution of GCs stimulated with low concentration (100 ng/mL RBP4); (**C**) flow cytometric analysis of the cell cycle distribution of GCs stimulated with medium concentration (200 ng/mL RBP4); (**D**) flow cytometric analysis of the cell cycle distribution of GCs stimulated with high concentration (200 ng/mL RBP4); (**E**) bar graph analysis of cell cycle distribution; (**F**) protein expression levels of CDK1, CCNE2, CCND1, and CCNA1; (**G**) immunoblotting results for CDK1, CCNE2, CCND1, and CCNA1; (**H**) relative mRNA expression levels of CDK1, CCNE2, CCND1, and CCNA1; (**I**–**L**) representative images of apoptosis; (**M**) quantitative analysis of cell apoptosis rate; (**N**,**O**) immunoblotting results and protein expression levels of Bax and Caspase-9; (**P**) relative mRNA content of Bax and Caspase-9; (**Q**) relative mRNA expression levels of StAR, CYP27B1, HSD3B1, and CYP17A1; (**R**) levels of E_2_ in cell supernatant; (**S**) levels of P_4_ in cell supernatant; (**T**,**U**) immunoblotting results and protein expression levels of StAR, CYP27B1, HSD3B1, and CYP17A1. ** indicates extreme significance (*p* < 0.01); * indicates significance (*p* < 0.05).

**Figure 4 animals-14-03118-f004:**
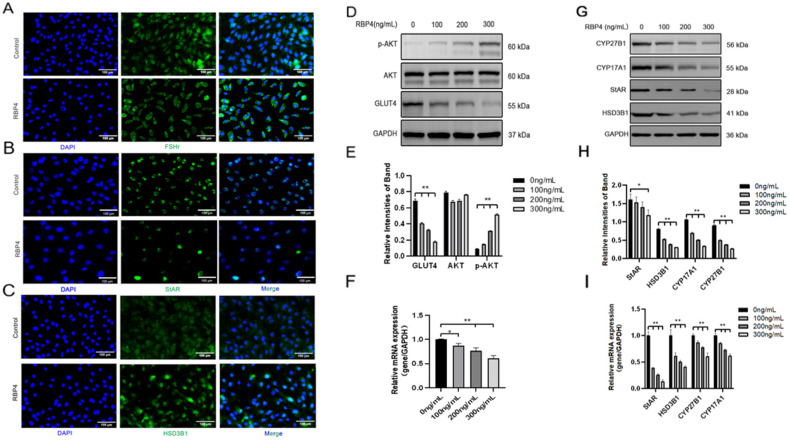
Exogenous RBP4 inhibits the expression of steroid hormones and GLUT4 in GCs. (**A**) Immunofluorescence staining of FSHR expression in GCs; (**B**) immunofluorescence staining of StAR expression in GCs; (**C**) immunofluorescence staining of HSD3B1 expression in GCs; (**D**) immunoblot results for AKT, p-AKT, and GLUT4; (**E**) protein expression levels of AKT, p-AKT, and GLUT4; (**F**) relative mRNA levels of GLUT4; (**G**) immunoblot results for CYP27B1, CYP17A1, StAR, and HSD3B1; (**H**) protein expression levels of CYP27B1, CYP17A1, StAR, and HSD3B1; (**I**) relative mRNA expression levels of StAR, CYP27B1, HSD3B1, and CYP17A1; ** indicates extreme significance (*p* < 0.01); * indicates significance (*p* < 0.05).

**Figure 5 animals-14-03118-f005:**
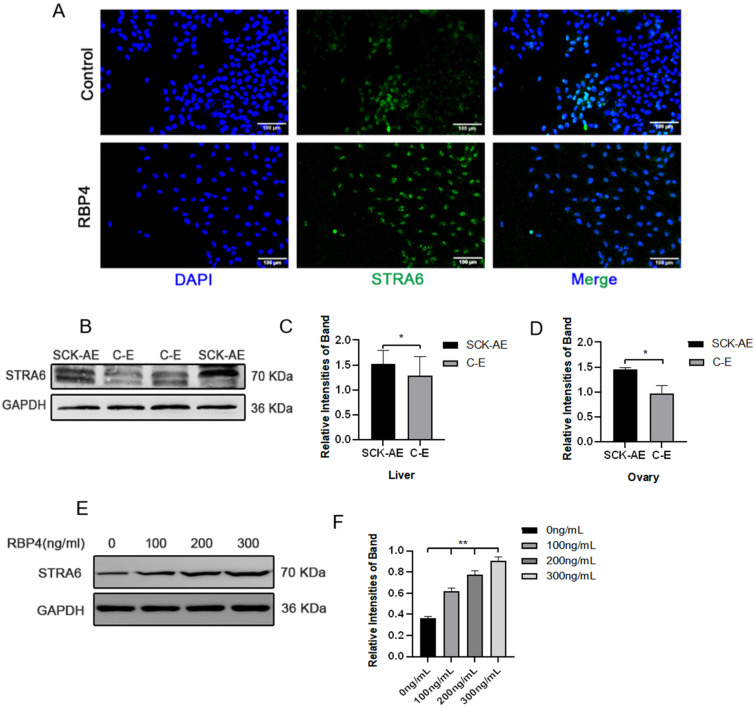
Expression of STRA6 in the bovine system. (**A**) Immunofluorescence of STRA6 expression in GCs under low-glucose culture conditions with or without added RBP4; (**B**) Western blot results for STRA6 expression in the liver and ovaries of SCK-AE and C-E group cows; (**C**) protein expression levels of STRA6 in the liver; (**D**) protein expression levels of STRA6 in the ovary; (**E**) immunoblotting results of STRA6 with increasing concentration of RBP4 addition in GCs; (**F**) protein expression levels of STRA6 in GCs. ** indicates extreme significance (*p* < 0.01); * indicates significance (*p* < 0.05).

**Figure 6 animals-14-03118-f006:**
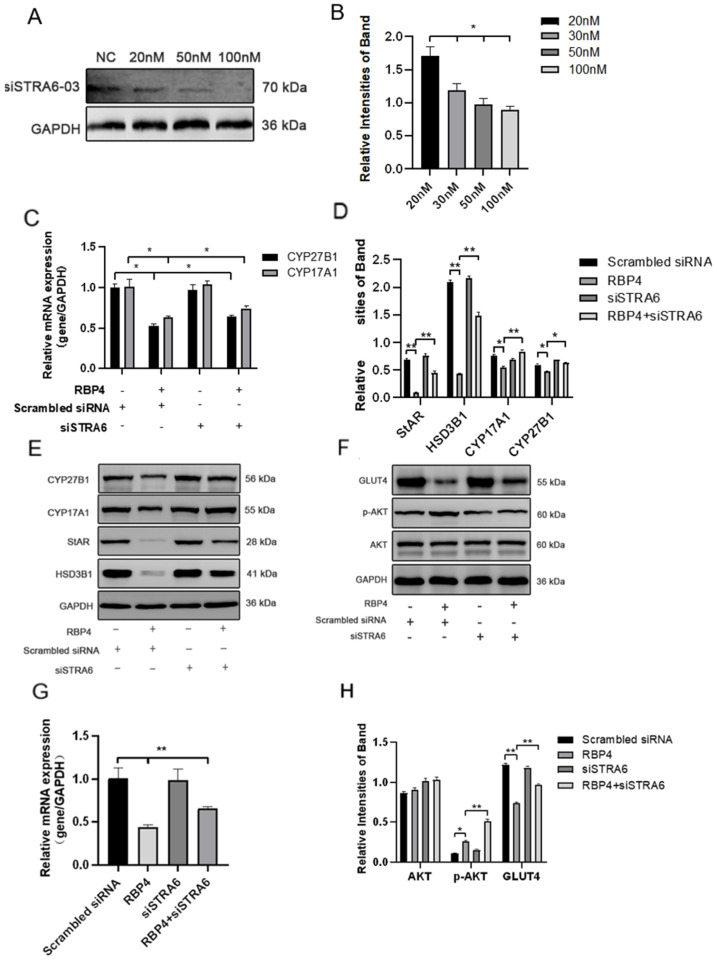
STRA6 silencing alleviates RBP4-induced suppression of steroid hormone secretion and GLUT4 expression. (**A**) Western blot results for siSTRA6 concentration screening; (**B**) protein expression levels at different siSTRA6 concentrations; (**C**) relative mRNA expression levels of CYP27B1 and CYP17A1; (**D**) protein expression levels of StAR, CYP27B1, HSD3B1, and CYP1A71; (**E**) Western blot results for StAR, CYP27B1, HSD3B1, and CYP1A71; (**F**) Western blot results for AKT, p-AKT, and GLUT4; (**G**) relative mRNA content of GLUT4; (**H**) protein expression levels of AKT, p-AKT, and GLUT4. ** indicates extreme significance (*p* < 0.01); * indicates significance (*p* < 0.05).

**Figure 7 animals-14-03118-f007:**
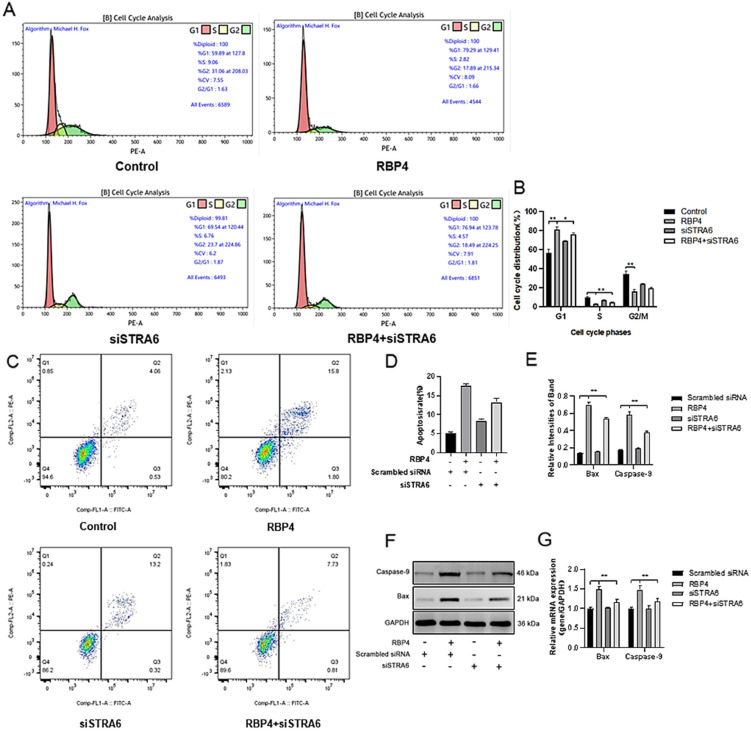
Alleviating RBP4-induced growth inhibition by STRA6 silencing. (**A**) Flow cytometry analysis of GCs’ cell cycle distribution; (**B**) bar graph analysis of cell cycle distribution at different phases; (**C**) representative images from flow cytometry; (**D**) quantitative analysis of apoptosis rate; (**E**) Western blot results for Bax and Caspase-9; (**F**) protein expression levels of Bax and Caspase-9; (**G**) relative mRNA content of Bax and Caspase-9. ** indicates extreme significance (*p* < 0.01); * indicates significance (*p* < 0.05).

**Figure 8 animals-14-03118-f008:**
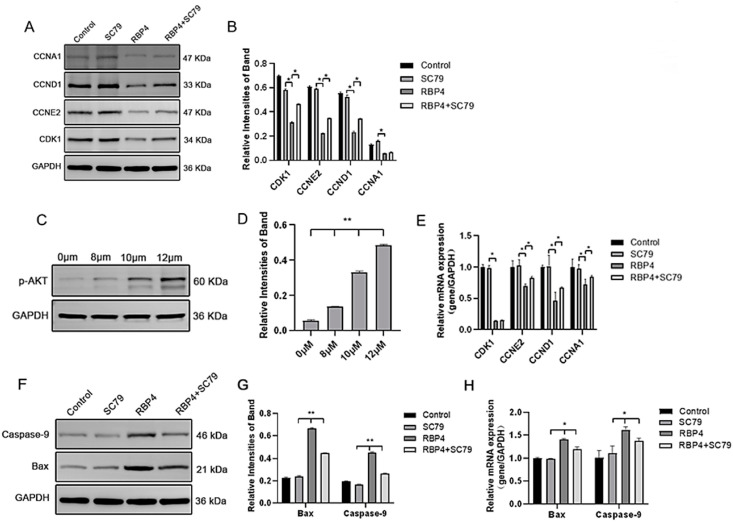
RBP4 induces apoptosis and suppresses GC proliferation via the PI3K/AKT pathway. (**A**) immunoblot results for CDK1, CCNE2, CCND1, and CCNA1; (**B**) protein expression levels of CDK1, CCNE2, CCND1, and CCNA1; (**C**) immunoblot results for SC79 concentration selection; (**D**) protein expression levels of p-AKT; (**E**) relative mRNA expression levels of CDK1, CCNE2, CCND1, and CCNA1; (**F**) immunoblot results for Bax and Caspase-9; (**G**) protein expression levels of Bax and Caspase-9; (**H**) relative mRNA levels of Bax and Caspase-9. ** indicates extreme significance (*p* < 0.01); * indicates significance (*p* < 0.05).

**Figure 9 animals-14-03118-f009:**
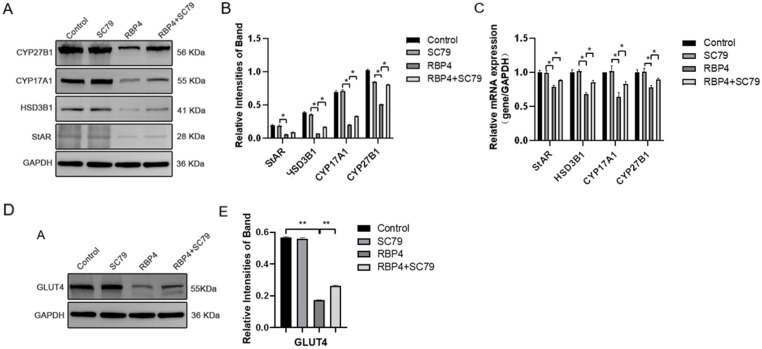
RBP4 inhibits steroid hormone protein expression via the PI3K/AKT pathway. (**A**) Immunoblot results for StAR, CYP27B1, HSD3B1, and CYP1A1; (**B**) levels of E_2_ in cell culture supernatant; (**C**) protein expression levels of StAR, CYP27B1, HSD3B1, and CYP1A1; (**D**) relative mRNA expression of CYP27B1, CYP17A1, StAR, and HSD3B1; (**D**) immunoblot results for GLUT4; (**E**) protein expression levels of GLUT4. ** indicates extreme significance (*p* < 0.01); * indicates significance (*p* < 0.05).

**Figure 10 animals-14-03118-f010:**
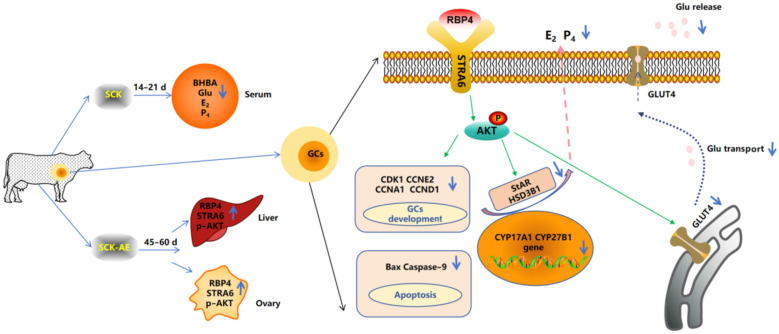
Mechanism diagram of RBP4-activated STRA6 inhibition of GCs’ activity and function through the PI3K/AKT signaling pathway.

**Table 1 animals-14-03118-t001:** Primer sequences used in qRT-PCR.

Gene	Gene ID	Amplicon Size (bp)	Forward Primer (5′→3′)	TM (°C)	Reverse Primer (5′→3′)	TM (°C)
β-actin	280979	132	CCTGGAGAAGAGCTACGAG	56.36	AAGGTAGTTTCGTGAATGCC	56.05
GAPDH	281181	197	TCTTCACTACCATGGAGAAGG	56.45	TCATGGATGACCTTGGCCAG	59.74
CDK1	281061	112	TGGCCAGAAGTGGAATCTTT	57.02	AGAGCAGATCCAAGCCATTT	57.18
CCNE2	538436	191	CGGGTCTGGCGAGGTTT	59.26	ACTGATGTTTCTTGGTGACCT	42.86
CCND1	524530	119	TACACTGACAACTCCATCCG	56.97	GAGAGGAAGTGCTCGATGAA	57.05
CCNA1	521939	156	AGTGAGAAGAGCGAGCAGTAA	58.84	TACTGTTCTTTGTGGCGGA	56.60
StAR	281507	86	AGGTGTGATAGCATGAGAGC	57.09	GCCAGATAACCCCATCTCAA	56.97
HSD3B1	281824	151	AGAGGATCATCTGCCTGTTG	57.00	TGCTCATCCAGAATGTCTCC	57.00
CYP17A1	281739	105	TCAAGGTGAAGATCGAGGTG	56.96	CTCCTCTAATTCTGTGCGGT	50.00
CYP27B1	539630	121	TCTGGTTAGAGAGACCATAGC	55.98	GAAGGGCTAAGCAGGTTAAT	55.10
GLUT4	282359	176	TGGAGTACTTAGGGCCAGAT	57.13	CAGCTTTCCCAAATCCCAAC	57.25
Bax	280730	179	AGCAGATCATGAAGACAGGG	57.00	CAGCTCCATGTTACTGTCCA	57.23
Caspase-9	100140945	195	CCGACATGATCGAGGACATT	57.48	ATGGGTCATCCTGTTTTGCT	57.39

## Data Availability

All data generated or analyzed during this study are included in the published article.

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
