# Peer review of "Elevated RBP4 in Subclinical Ketosis Cows Inhibits Follicular Granulosa Cell Proliferation and Steroid Hormone Synthesis"

_animals, 2024, doi:10.3390/ani14213118_

Round 1
Reviewer 1 Report
Comments and Suggestions for Authors
The manuscript needs revision. Please refer to comments given in the text of reviewed attached file of the manuscript.

Author Response
1.Question: What is the basic problem that your research focuses on and is done to solve?Please specify the main knowledge gap that your article has filled in the text.
Answer: We have rewritten the first sentence of the abstract, describing the problem that the experiment will solve.
2.Question: The conclusion is very wide and general.It is better to add a specific conclusion from your specific results.
Answer: Thank you for your suggestion. Glucose metabolism encompasses a wide range of processes, and we only tested GLUT4, it is insufficient to fully represent cellular glucose metabolism.. Therefore, we have revised the title, conclusion and removed the content related to glucose metabolism from the manuscript (in Line 30-32).
3.Question: It is better to explain about importance and application of animal breeding, especially cattle. For this you can use added sentences and references.
Answer: Thank you for providing the references, but we believe that the relevance between the references you provided and our research content is relatively low. Our research aims to address the mechanisms underlying postpartum reproductive dysfunction in dairy cows caused by energy metabolism disorders.
4.Question: Please specify in the objective whether your research is being conducted for the first time in the world or is it a continuation of another research?Please specify the main knowledge gap that your article has filled.
Answer: Thank you for your suggestion, it has been very helpful in improving the content of our manuscript. We have added a description of the purpose of this study in Line 96-99 & Line 101-104.
5.Question: please add full name for abbreviations in the first place of text.
Answer: Thank you, we have added the full name (Line 116).
6.Question: indices or indexes??? Please check.
Answer: We have changed indices to indices in Line 176.
7.Question: this sentence isn't complete and clear?! what do you mean? tablethis sentence isn't complete and clear?!?
Answer: We apologize for the writing error and have rewritten the sentence (in Line 237-240).
8.Question: identify quality and quantity of extracted RNA?
Answer: Dear reviewer, we measured the extracted RNA concentration (Line 233-237) using a NanoDrop 2000 spectrophotometer (Thermo Fisher Scientific, USA), and all RNA OD260/280 values were between 1.8 and 2.0, indicating that our RNA quality is suitable for subsequent experiments. The concentration of purified RNA was determined by UV spectrum at 260 nm.
9.Question: did you design primers or take from other references?
if you designed, please explain in the text how? and add your used software and ....
If you took from other references, please add reference in the text.
Answer: We designed our own genes, not based on references. We have added a description of the relevant methods for gene design in the manuscript (in Line 244-246).
10.Question: it is better to identify in the text reference genes, beta actin and GAPDH! and explain that did you use both reference genes for all genes or not??
Answer: All of our genetic experiments utilized two internal references, and we have added a description of this in the manuscript (in Line 243-244).
11.Question: please add accession number and Tm for used primers in the table!!
Answer: We have added accession number and Tm (Table 1).
12.Question: did you use animal statistical model? Please add your used model and its components in the text!
Answer: We did not employ animal statistical models. We randomly selected postpartum cows and divided them into experimental and control groups based on their serum BHBA and estrus status. On this basis, one-way ANOVA was applied to compare the levels of RBP4 and Glu in the serum of two groups of cows.
13.Question: Please compare your original result with the previous results and identify what interesting and new results you have added to the previous results.
Answer: We have rewritten this sentence based on your suggestion (Line 580-583).
14.Question: The conclusion is very wide and general. It is better to add a specific conclusion from your specific results.
Answer: Thank you for your suggestion. We have removed the relevant description of Glu metabolism from the conclusion.

Reviewer 2 Report
Comments and Suggestions for Authors
The manuscript submitted for review examined whether elevated retinol binding protein4 (RBP4) in cows with subclinical ketosis affects granulosa cell viability and steroid function and interferes with glucose utilization by the cells. The authors compared subclinical ketosis cows in their second to third month after parturition to comparable cows without subclinical ketosis. Through a progressive set of in vivo and in vitro cell culture studies, they determined that elevated RBP4 inhibited cell mitosis, promoted apoptosis, and interfered with glucose metabolism by inhibiting GLUT4 transport of glucose. The authors determined that RBP4 acts through its receptor STRA6 and the PI3/AKT pathway to exert its effects. The study's major caveat is that by pooling tissues, they did not consider cow variability. Overall, well-designed and executed studies leading to sound conclusions.
Comments on the Quality of English LanguageL 13 make into two sentences "…in vitro. The results"
L 14-18 Consider combining these two sentences – redundant wording
L 26 change are to were and is to was
L 28 phrase "were somewhat alleviated
L45 change of to by" detracts from the significance
L 40 change to "bodies mobilize stored fat"
L 103 lower case g add verb was - Animal grouping was … Agriculture. This
L 114 insert period after). Move “from day 14 to 21 postpartum” to follow mM
L 116 change divided into to assigned to
Insert and before BHBA
L 133 again at
L 135 combine sentences by removing “Electric shock” and “was performed”
L 140 “disinfected surgical knives were”
L 142 delete bilateral
L 173 delete were used
L 174 change were to was
l 187 Delete section – not used
l 207 Sections 2.7, 2.9, 2.11, and 2.12 need rewording to have subjects performing actions. These read as instructions for the lab manual.
L265 Delete “All experiments were repeated, and the average…”- these are replicates as noted subsequently. Change is to are.
L 274 measured
L 297 change are to were and can to could
L 310 Is this phrase supposed to follow influencing on l 207?
L 321 NEB, which is
L 340 capitalize These
Author Response
Reviewer 2
- Question: L 13 make into two sentences "…in vitro. The results".
Answer: Thank you, we have made modifications (in Line 12).
- Question: L 14-18 Consider combining these two sentences – redundant wording.
Answer: We have merged the two sentences into one sentence (in Line 12-14).
- Question: L 26 change are to were and is to was.
Answer: We have made modifications (in Line 22).
- Question: L 28 phrase "were somewhat alleviated
Answer: We have already deleted "somewhat" (in Line 24).
- Question: L 45 change of to by "detracts from the significance"
Answer: We have made modifications (in Line 41).
- Question: L 40 change to "bodies mobilize stored fat"
Answer: We have made modifications (in Line 36).
- Question: L 103 lower case g add verb was - Animal grouping was … Agriculture. This.
Answer: We have made modifications (in Line 104).
- Question: L 114 insert period after). Move “from day 14 to 21 postpartum” to follow mM.
Answer: We have made modifications (in Line 116).
- Question: L 116 change divided into to assigned to. Insert and before BHBA.
Answer: We have made modifications (in Line 116).
- Question: L 133 again at.
Answer: We have made modifications (in Line 135).
- Question: combine sentences by removing “Electric shock” and “was performed”
Answer: We have made modifications (in Line 138).
- Question: L 140 “disinfected surgical knives were”
Answer: We have made modifications (in Line 142).
- Question: L 142 delete bilateral.
Answer: We have deleted.
- Question: L 173 delete were used.
Answer: Thanks, we have deleted.
- Question: L 174 change were to was.
Answer: We have made modifications (in Line 176).
- Question: l 187 Delete section – not used.
Answer: Our experiment used Western Blot (WB), and we have revised the title of this section accordingly.
- Question:l 207 Sections 2.7, 2.9, 2.11, and 2.12 need rewording to have subjects performing actions. These read as instructions for the lab manual.
Answer: We have rewritten the content in sections 2.7, 2.9, 2.11, and 2.12.
- Question:L265 Delete “All experiments were repeated, and the average…”- these are replicates as noted subsequently. Change is to are.
Answer: We have made modifications (in Line 268).
- Question:L 274 measured.
Answer: We have made modifications (in Line 277).
- Question:L 297 change are to were and can to could.
Answer: We have made modifications (in Line 301).
- Question:L 310 Is this phrase supposed to follow influencing on l 207?
Answer: We believe that this sentence should still be placed in the results section.
- Question:L 321 NEB, which is. L 340 capitalize These.
Answer: Dear Reviewer, we were unable to locate the word you mentioned in the corresponding line. We have checked for spelling and capitalization issues throughout the text and made revisions accordingly. Finally, we sincerely thank you for all the suggestions, which have greatly improved the quality of the manuscrip

Reviewer 3 Report
Comments and Suggestions for Authors
The manuscript is well designed and performed. Several comments required to be answered.
1. The references should be updated.
2. The replicate times shoud be addressed.
3. The quality of WB picutres should be improved.
Comments on the Quality of English LanguageThe language of the paper should be polished by native english speakers.
Author Response
- Question: The references should be updated.
Answer: Thank you for your suggestion. We have replaced some outdated references.
- Question: The replicate times should be addressed.
Answer: We conducted three parallel experiments in all of our in vitro experiments.
- Question:The quality of WB pictures should be improved.
Answer: Dear reviewer, all WB images are already presented at their highest resolution. Additionally, we have provided the original images for the readers' reference.
- Question:The language of the paper should be polished by native English speakers.
Answer: We have corrected the spelling and grammar errors in the manuscript. All revisions content has been highlighted in yellow.